# Apicortin defines the *Plasmodium* apical conoid body but is dispensable for the parasite life cycle

Mohammad Zeeshan[1,2], Akancha Mishra[1], Sarah L Pashley[1], Robert Markus[1], Declan Brady[1], Anthony A Holder[3], Carolyn Moores[4], Rita Tewari[1]

Apicomplexan parasites such as *Plasmodium* spp. and *Toxoplasma gondii* possess unique tubulin-based structures, including subpellicular microtubules and apical polar rings, which are essential for parasite motility, host cell invasion, and replication. Apicortin, a microtubule-associated protein, contains a doublecortin (DC) domain and a partial tubulin polymerization-promoting protein (TPPP) domain, both implicated in microtubule binding and stabilization. How tubulin-based structures are maintained is poorly understood, but it may involve Apicortin, so far found only in apicomplexans and the placozoan *Trichoplax adhaerens*. Here, we investigated the location and function of Apicortin in *Plasmodium berghei*. Live-cell imaging of a transgenic parasite line expressing GFP-tagged Apicortin showed its location at the apical end of invasive parasite stages within the mosquito vector. Super-resolution and expansion microscopy revealed that Apicortin forms a distinct ringlike structure in the apical complex region at the apical end. However, deletion of the Apicortin gene had no effect on parasite development, indicating that this protein is not essential. This suggests that there may be redundancy or compensatory functions in the mechanisms that stabilize the apical complex.

## Introduction

*Plasmodium*, the causative agent of malaria, is the most dangerous vector-borne protozoan parasite, responsible for an estimated 597,000 deaths in 2023 (WHO, 2024). It belongs to the phylum Apicomplexa, a group of obligate intracellular parasites, and is characterized by invasive stages—the merozoite, ookinete, and sporozoite—that are highly polarized cells. A defining feature of these cells is the presence of an anterior apical complex, which has a crucial role in parasite motility and invasion of host cells and tissues. Unlike many eukaryotic cells that use cilia or flagella for motility, apicomplexan parasites use a unique actomyosin motor known as the glideosome. This motor is located within the pellicle, a membranous structure comprised of the plasma membrane and the inner membrane complex, and drives the gliding motility essential for host cell invasion (Frenal et al, 2017).

The apical complex has several distinct structures, including an apical cap with polar rings, a subpellicular microtubule-organizing centre (MTOC), and secretory organelles including micronemes, rhoptries, and dense granules, all located on the cytosolic side of the apical complex. Another important component is the conoid—a specialized, cone-shaped structure. In some apicomplexan species, such as *Toxoplasma gondii* and *Eimeria* spp., the conoid is well defined and functional, but in other species like *Plasmodium* spp., there are only remnants of the conoid, probably reflecting a divergent adaptation of this structure (Dos Santos Pacheco et al, 2020; Koreny et al, 2021; Zeng et al, 2025).

The apical conoid is found predominantly in apicomplexans, particularly within the coccidia, and has structural similarity to the pseudoconoid of related alveolate lineages such as *Chromera* and *Perkinsus*. In these non-apicomplexans, the pseudoconoid is closely associated with the flagellar root apparatus, suggesting a common evolutionary relationship between apical complex components and flagellar structures that may have been retained or repurposed in different lineages (Portman & Slapeta, 2014).

Our recent investigation of the conoid complex revealed that many of the apical complex proteins are conserved across apicomplexan parasites (Koreny et al, 2021). Among these proteins is Apicortin (also referred to as doublecortin), which contains domains characteristic of both tubulin polymerization-promoting proteins (TPPP) and the doublecortin (DC) family (Orosz, 2016, 2021). Although both the TPPP and DC domains are widely distributed in metazoan proteins, the presence of both domains within a single protein is an almost exclusive feature of Apicortin. This distinctive domain architecture likely reflects a specialized adaptation of the parasite's cytoskeleton to the requirements of a highly polarized invasive cell (Orosz, 2009, 2016).

[1]School of Life Sciences, University of Nottingham, Nottingham, UK  [2]Division of Molecular Microbiology and Immunology, CSIR-Central Drug Research Institute, Lucknow, India  [3]Malaria Parasitology Laboratory, The Francis Crick Institute, London, UK  [4]Institute of Structural and Molecular Biology, School of Natural Sciences, Birkbeck, University of London, London, UK

Correspondence: rita.tewari@nottingham.ac.uk

Elegant studies have described the molecular architecture of Apicortin and its role in *T. gondii* tachyzoite invasion of host cells (Nagayasu et al, 2017; Leung et al, 2020). These investigations demonstrated that the absence of Apicortin severely impairs both host cell invasion and parasite replication (Nagayasu et al, 2017). Structural analyses determined the three-dimensional structure of the DC domain, and heterologously expressed Apicortin promoted the formation of microtubule (MT) bundles and short, strongly curved, arc-like structures (Leung et al, 2020).

*Plasmodium* species lack a defined conoid structure, but they retain a functional apical complex and encode an Apicortin homologue. However, the subcellular location and function of *Plasmodium* Apicortin are largely unknown. A recent study reported Apicortin expression in *Plasmodium falciparum* asexual blood stages, where it interacts with both α- and β-tubulins, and its repression leads to impaired host cell invasion and parasite growth (Chakrabarti et al, 2020, 2021). However, its location and function in the stages of the life cycle in the mosquito remain uncharacterized.

Here, we report on the expression dynamics and function of Apicortin in *Plasmodium berghei*, a rodent malaria model which enables the investigation of all invasive stages in the life cycle—merozoites, ookinetes, and sporozoites. We demonstrate that the invasive forms (ookinetes and sporozoites) that are present only within the mosquito vector, express Apicortin, whereas merozoites do not. Apicortin is associated with the apical complex, forming a distinct ringlike structure. Despite this specific location of the protein, deletion of the *apicortin* gene had no effect on parasite development and progression through the life cycle.

## Results

### Apicomplexan parasites have Apicortins with conserved TPPP and DC domains

We analysed the structure of Apicortin in *Plasmodium* and other apicomplexans using a range of bioinformatics tools including InterPro, Clustal Omega, and AlphaFold. *P. berghei* Apicortin has a TPPP and a DC domain (Fig 1A), and there is moderate sequence homology between *P. berghei*, *P. falciparum*, and *T. gondii* Apicortins (Fig 1B). Apicortins display a distinctive domain arrangement and overall limited sequence identity (Fig 1C), including within the structurally well-defined DC domain (Fig 1D).

### *Pb*Apicortin is located at the apical end of invasive parasites in the mosquito vector

To investigate the expression and location of *Plasmodium* Apicortin, we generated a transgenic *P. berghei* line expressing Apicortin with a C-terminal GFP tag. An in-frame *gfp* coding sequence was inserted at the 3′ end of the endogenous *apicortin* locus using single-crossover homologous recombination (Fig S1A). Successful insertion was confirmed by diagnostic PCR (Fig S1B).

Both asexual blood stages in the mammalian host and sexual stages in the mosquito of this PbApicortin-GFP transgenic parasite

line were examined for the expression of Apicortin, using live-cell imaging. Expression was undetectable during asexual blood-stage replication, in both male and female gametocytes, and in flagellated male gametes. PbApicortin-GFP was first detected at the zygote stage, ~2 h after fertilization, located at the cell periphery as a single focal point (Fig 2A). To follow the location of PbApicortin-GFP during zygote differentiation, we examined the different stages of zygote development (stages I–V) to the fully motile and invasive ookinete (stage VI) (Fig 2A). PbApicortin-GFP was detected at a single focal point during stage 1 on the cellular protrusion defining the apical polarity that develops as the zygote starts to transform into a zoite. As the zoite elongated through stages II to V, PbApicortin-GFP remained located at the polar end of the protrusion and eventually at the apical end of the mature ookinete (stage VI) (Fig 2A). To assess the effect of microtubule-stabilizing (Taxol) and microtubule-destabilizing (nocodazole) compounds on PbApicortin-GFP localization, we treated zygotes of the PbApicortin-GFP line with nocodazole or Taxol and examined GFP distribution in mature ookinetes (Fig S1C). Nocodazole treatment did not affect ookinete conversion, morphology, or Apicortin-GFP localization (Fig S1D–F). Taxol treatment impaired ookinete shape transformation; however, Apicortin-GFP remained localized at the apical end (Fig S1D–F).

To compare the location of PbApicortin-GFP relative to that of *P. berghei* end-binding microtubule protein 1 (PbEB1), a protein known to be located at the apical end of the ookinete, we genetically crossed PbApicortin-GFP (green) and PbEB1-mCherry (red) parasite lines to obtain parasites expressing both fluorescent markers. *Pb*EB1 is located at the apical end of the cell during ookinete development (Zeeshan et al, 2023) and has been shown to exhibit lateral attachment to spindle MTs in the nucleus (Yang et al, 2023; Zeeshan et al, 2023). Live-cell imaging of these parasites showed that the location of Apicortin (green) is more apical than that of PbEB1 (magenta) during ookinete development (stages I–IV) (Fig 2B). In later more mature ookinete stages, the apical PbEB1-mCherry signal becomes progressively diffuse and finally disappears, but the apical PbApicortin-GFP signal remains until the end of ookinete development (Fig 2B).

The mature ookinete transverses the mosquito gut wall to form an oocyst on the outer surface of the gut wall, where multiple nuclear divisions produce thousands of sporozoites (Guttery et al, 2022). Therefore, we investigated the expression of Apicortin during this oocyst development. We observed a diffuse location of PbApicortin-GFP in oocysts with some distinct foci (Fig 2C). The sporozoites that emerged from these oocysts at day 14 after mosquito infection had a strong apical focal point of PbApicortin-GFP expression, together with a diffused distribution within the sporozoite (Fig 2D). A similar pattern of PbApicortin-GFP distribution was seen in salivary gland sporozoites.

### Ultrastructural analysis of *Pb*Apicortin shows a ringlike structure at the apical end of ookinetes and sporozoites

To further examine the location of Apicortin, structured illumination microscopy (3D-SIM) was performed on fixed developing ookinetes (stage II) and fully mature ookinetes (stage VI). We found that the focal point of PbApicortin-GFP observed in wide-field

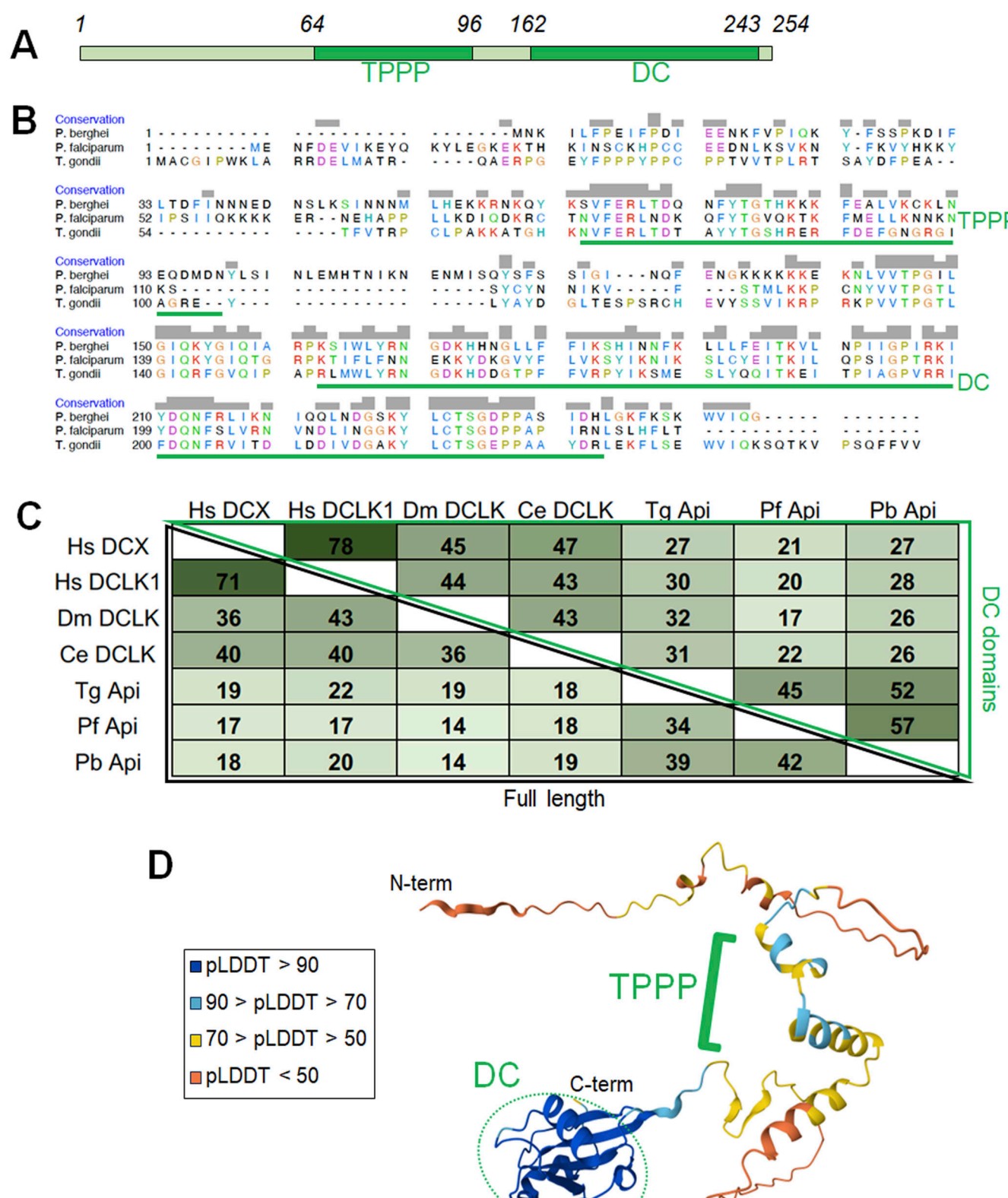

**Figure 1. Sequence and structural analysis of Apicortins.**
**(A)** Schematic of *P. berghei* Apicortin (PBANKA_1232600; UniProt: A0A509AMB2) with residue numbers indicating UniProt domain boundaries. TPPP: Pfam PF05517; DC: Pfam PF03607. **(B)** Clustal Omega protein sequence alignment of *P. berghei*, *P. falciparum* (UniProt: C0H4E4), and *T. gondii* (UniProt: A0A7J6K3U3) Apicortin with putative microtubule-binding domains indicated. **(C)** Sequence conservation of selected DC-containing proteins with full-length sequence similarity depicted on the left of the matrix (black outline) and DC domain similarity on the right (green outline). Hs DCX, *H. sapiens* doublecortin, UniProt: O43602; Hs DCLK1, *H. sapiens* doublecortin-like

microscopy is a ringlike structure at the apical end of the ookinete protrusion (Fig 3A), suggesting that Apicortin is part of a conoid remnant, consistent with our previous prediction (Koreny et al, 2021).

To define in more detail the ultrastructure, we analysed the location of PbApicortin-GFP relative to subpellicular MTs using ultrastructure expansion microscopy (U-ExM) analysis of ooki-netes. Subpellicular MTs are organized at apical ring 2 (APR2) (Koreny et al, 2021) in *Plasmodium* and play a crucial role in de-termining the shape, size, and polarity of the parasite cell. U-ExM images revealed Apicortin as a ringlike structure at the apical end of the ookinete (Fig 3B), consistent with the 3D-SIM observation (Fig 3A). Subpellicular MTs stained with antibody to alpha-tubulin were visible appearing from the Apicortin ring (Fig 3B).

Together, live-cell imaging and ultrastructural analysis of ookinetes indicate that Apicortin forms a ringlike structure at the apical end of the cell, distal to EB1 and connected to the sub-pellicular MTs, as summarized in the schematic (Fig 3C). Previously, we had proposed that Apicortin is part of the conoid body (Koreny et al, 2021) and our current finding supports this suggestion. The schematic (Fig 3C) shows the arrangement of the apical complex and its different structures, with the possible location of Apicortin highlighted, based on a previous prediction and current findings with ookinetes.

To examine in more detail the location of Apicortin in sporo-zoites within the mosquito, 3D-SIM and expansion microscopy were used. A similar location was observed by both 3D-SIM (Fig 3D) and U-ExM (Fig 3E) in sporozoites, consistent with the proposed location of Apicortin at the apical ring of these invasive stages.

### Apicortin is not necessary for parasite transmission by mosquitoes

The function of Apicortin during the *Plasmodium* life cycle was examined by deleting its gene from the *P. berghei* genome. This was achieved using a double-crossover homologous recombina-tion strategy, using a parasite line that constitutively expresses GFP throughout the parasite life cycle (Fig S2A) (Janse et al, 2006). The successful integration of the targeting construct at the *api-cortin* locus was confirmed by integration PCR (Fig S2B). Successful creation of this transgenic parasite (*ΔPbApicortin*) indicated that the gene is not essential for the asexual blood stages. Further phenotypic analysis of the *ΔPbApicortin* parasite compared with the parental parasite (WT-GFP) was carried out at other stages of the life cycle.

*ΔPbApicortin* and WT-GFP parasites produced a comparable number of gametocytes in mice, so we next analysed male and female gametocyte differentiation. Male gamete development in *Plasmodium* is a rapid process with three rounds of genome duplication and axoneme formation resulting in eight motile flagellated gametes, in a process named exflagellation (Sinden et al, 1976). There was no defect in male gamete exflagellation and

female gamete formation in *ΔPbApicortin* parasite compared with WT-GFP (Fig 4A). Zygote formation and ookinete differentiation were also similar in *ΔPbApicortin* and WT-GFP parasites (Fig 4B). The shape and motility of ookinetes were also not affected by the *apicortin* gene deletion (Fig 4C and D, Video 1 and Video 2). Detailed structural analysis by U-ExM revealed no morphological differ-ences in apical polarity and MT organization in *ΔPbApicortin* and WT-GFP ookinetes (Fig 4E). We counted the NHS ester–stained dense granules that may correspond to micronemes and apical vesicles but observed no significant difference (Fig S2C). To assess the sensitivity of *ΔPbApicortin* to the effects of Taxol and noco-dazole, we treated zygotes of the *ΔPbApicortin* line with these compounds and examined the ookinete conversion and mor-phology of mature ookinetes. Nocodazole treatment did not affect ookinete conversion and morphology (Fig S2D and E). Taxol treatment impaired ookinete shape transformation in both WT-GFP and *ΔPbApicortin*, suggesting that the effect is due to Taxol treatment alone and is not affected by the Apicortin deletion (Fig S2D and E).

To investigate any loss of function in oocyst development and sporogony resulting from deletion of the *apicortin* gene, *Anopheles stephensi* mosquitoes were fed on mice infected with *ΔPbApicortin* parasites or WT-GFP parasites as a control. The number of GFP-positive oocysts on the mosquito gut wall was counted on days 14 and 21 post-infection. There was no significant difference in the number of *ΔPbApicortin* and WT-GFP oocysts (Fig 4F), and the size of the oocysts was similar for both parasites (Fig 4G). There was no significant difference in *ΔPbApicortin* sporozoite numbers in oocysts at days 14 and 21 post-infection, or in salivary glands at day 21 compared with WT-GFP parasites (Fig 4H and I). The shape, size, and motility of salivary gland sporozoites were indistinguishable from those of WT-GFP parasites (Fig 4J and K, Video 3 and Video 4). When infected mosquitoes were used in bite-back experiments to ascertain the infectivity of *ΔPbApicortin* sporozoites in mice, a blood-stage infection was observed after 4 d with both *ΔPbApicortin* and WT-GFP sporozoites (Fig 4L), indicating that Apicortin is not necessary for parasite transmission through the mosquito.

## Discussion

In this study, we provide new insights into the sequence con-servation, structural features, and functional relevance of Api-cortins in *Plasmodium* and other apicomplexan parasites. Through a combination of comparative sequence analysis and AlphaFold-based structural modelling, we show that *Plasmodium* Apicortins retain a conserved architecture featuring TPPP and DC domains, consistent with the microtubule/tubulin-associated role of Api-cortin in the *T. gondii* tachyzoite apical conoid structure (Nagayasu et al, 2017; Leung et al, 2020; Sun et al, 2022). Both TPPP and DC domains are microtubule or tubulin polymer-binding modules (Orosz, 2021), and despite only moderate sequence conservation,

kinase 1, UniProt: O15075; Dm DCLK, *D. melanogaster* doublecortin-like kinase, UniProt: Q7PL17; Ce DCLK *C. elegans* doublecortin-like kinase, UniProt: Q95QC4. **(D)** AlphaFold predicted structure of *P. berghei* Apicortin, coloured according to the per-residue confidence metric, pLDDT from 1 to 100, and with TPPP and DC domains and N and C termini indicated.

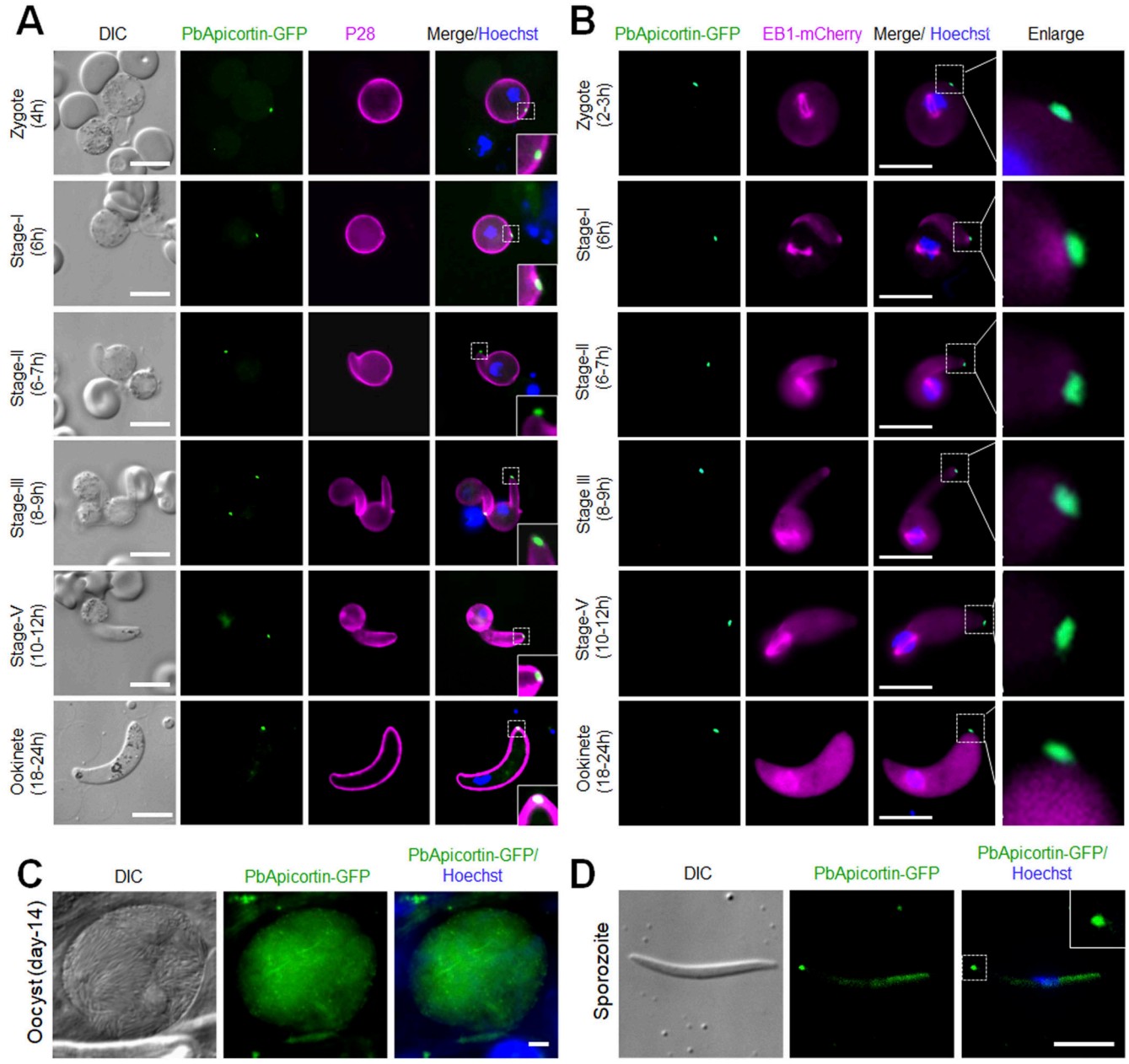

**Figure 2.** *P. berghei* **Apicortin has an apical location during zygote transformation to ookinete and during sporogony.**
**(A)** Live-cell imaging showing *Pb*Apicortin-GFP location during zygote-to-ookinete transformation. A cy3-conjugated antibody, 13.1, which binds to the P28 protein on the surface of activated female gametes, zygotes, and ookinetes, was used to mark these stages (magenta). Panels: DIC (differential interference contrast); PbApicortin-GFP (green, GFP); merged: Hoechst (blue, DNA), PbApicortin-GFP (green, GFP), and P28 (magenta). Scale bar = 5 μm. Insets show the zoom of PbApicortin-GFP signal. **(B)** Live-cell imaging showing the location of PbApicortin-GFP (green) with respect to EB1-mCherry (magenta) during zygote-to-ookinete transformation. Scale bar = 5 μm. **(C)** Live-cell imaging of PbApicortin-GFP in oocysts in mosquito guts at 14 d post-infection Panels: DIC (differential interference contrast); Apicortin-GFP (green, GFP); merged: Hoechst (blue, DNA) and PbApicortin-GFP (green, GFP). Scale bar = 5 μm. **(D)** Live-cell imaging of PbApicortin-GFP in sporozoites. Panels: DIC (differential interference contrast); PbApicortin-GFP (green, GFP); merged: Hoechst (blue, DNA) and PbApicortin-GFP (green, GFP). Scale bar = 5 μm.

particularly within the DC domain, the maintenance of this distinct domain combination in apicomplexans suggests a specialized role of Apicortin in the unique conoid structures of these parasites.

We demonstrated the location of Apicortin at the apical end of ookinetes and sporozoites as a ringlike structure during these invasive parasite stages in the mosquito. In ookinetes, the apical location is established early during zygote development and persists through ookinete maturation. Its location is maintained in both mid-gut and salivary gland sporozoites. The effects of Taxol and the microtubule destabilizer, nocodazole, were not modulated by the presence or absence of Apicortin, and no effect on ookinete conversion was observed.

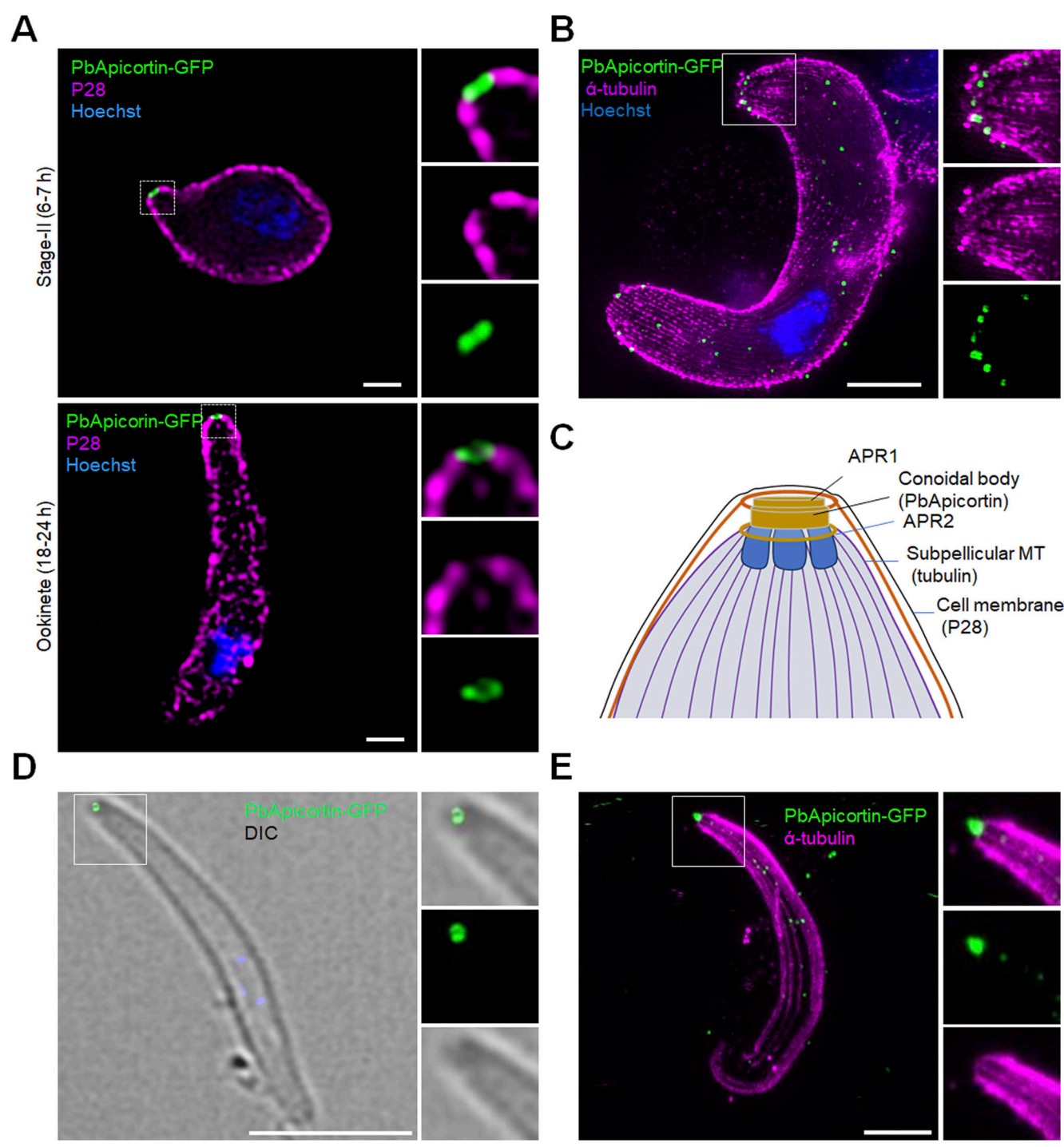

**Figure 3.  Ultrastructural analysis reveals a PbApicortin ringlike structure associated with subpellicular microtubules.**
**(A)** 3D-SIM images showing PbApicortin-GFP location during ookinete differentiation (stages II and VI). Hoechst (blue, DNA), PbApicortin-GFP (green, GFP), and P28 (magenta). Scale bar = 1 μm. **(B)** U-ExM image showing ringlike structure of PbApicortin-GFP (green) and subpellicular MTs (magenta) stained with alpha-tubulin antibody. Scale bar = 5 μm. **(C)** Schematic of the proposed apical region of an ookinete showing the location of Apicortin. **(D)** 3D-SIM images showing PbApicortin-GFP location in sporozoites. Hoechst (blue, DNA), PbApicortin-GFP (green, GFP). Scale bar = 5 μm. **(E)** U-ExM image of sporozoite showing the ringlike structure of PbApicortin (green) and subpellicular MTs (magenta) stained with alpha-tubulin antibody. Scale bar = 5 μm.

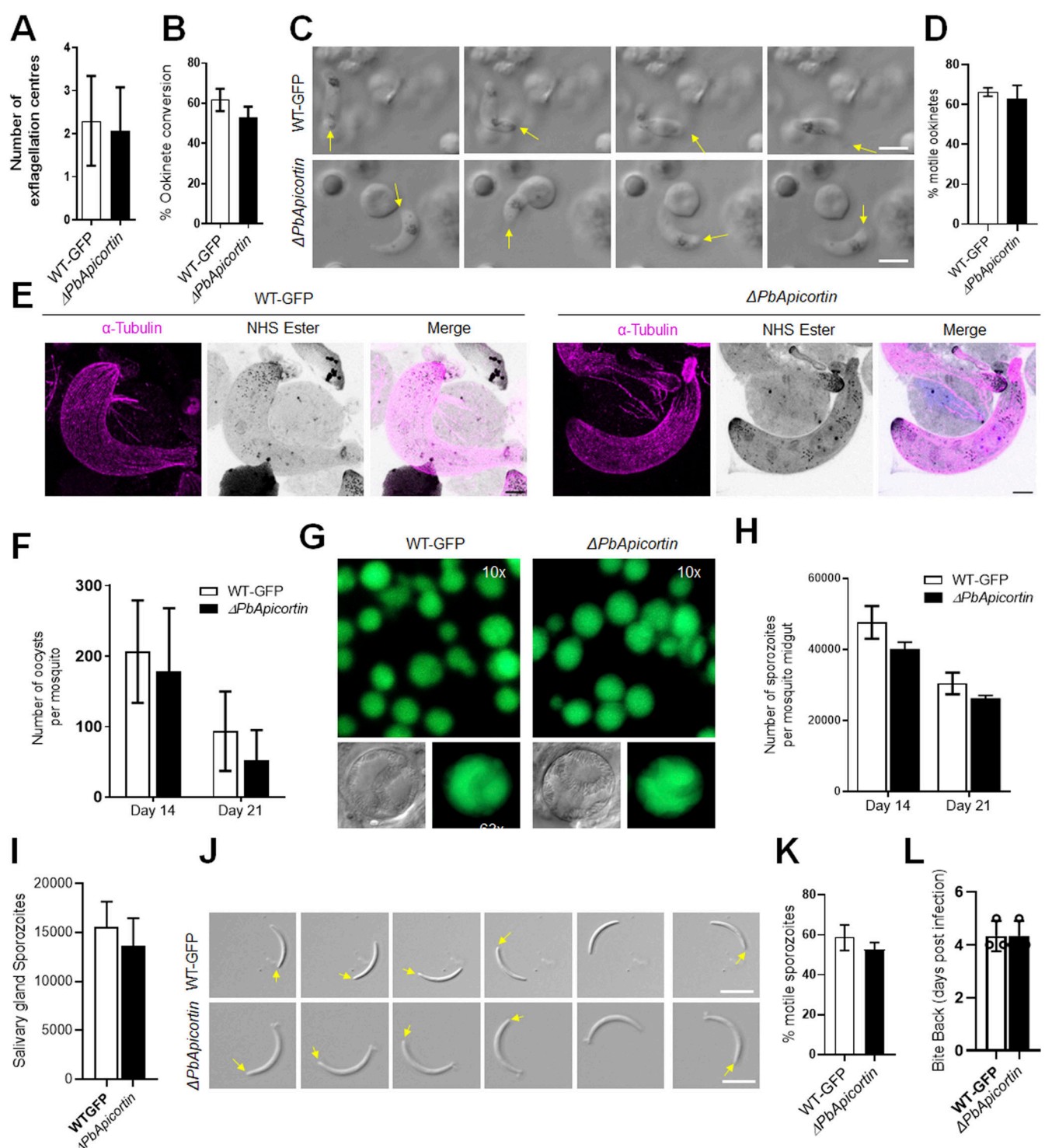

**Figure 4. PbApicortin is not necessary for parasite transmission.**
**(A)** Male gametogony (exflagellation) of *ΔPbApicortin* (black bar) and WT-GFP (white bar) lines measured as the number of exflagellation centres per field. Mean ± SD; n = 3 independent experiments. **(B)** Ookinete conversion as a percentage for *ΔPbApicortin* (black bar) and WT-GFP (white bar) parasites. Ookinetes were identified using 13.1 antibody as a surface marker and defined as those cells that differentiated successfully into elongated "banana-shaped" ookinetes. Mean ± SD; n = 3 independent experiments. **(C)** Differential interference contrast (DIC) time-lapse image sequences showing motile *ΔPbApicortin* and WT-GFP ookinetes. **(D)** Quantitative data for motile ookinetes for *ΔPbApicortin* and WT-GFP based on two independent experiments. **(E)** U-ExM images showing ookinetes labelled with anti-tubulin antibody and NHS ester. Scale bar = 5 μm. **(F)** Total number of GFP-positive oocysts per infected mosquito in *ΔPbApicortin* (black bar) and WT-GFP (white bar) parasites at 14 and 21 d post-infection (dpi). Mean ± SD; n = 3 independent experiments. **(G)** Mosquito mid-guts at 10x and 63x magnification showing oocysts of *ΔPbApicortin* and WT-GFP lines at 14 dpi. Scale bar = 50 μm in 10x and 20 μm in 63x. **(H)** Total number of sporozoites in oocysts of *ΔPbApicortin* (black bar) and WT-GFP (white bar) parasites at 14 and 21 dpi. Mean ± SD; n = 3 independent experiments. **(I)** Total number of sporozoites in salivary glands of *ΔPbApicortin* (black bar) and WT-GFP (white bar) parasites. Bar diagram shows the mean ± SD; n = 3 independent experiments. **(J)** Differential interference contrast (DIC) time-lapse image sequences showing motile *ΔPbApicortin* and

Interestingly, we did not detect Apicortin expression in the asexual erythrocytic stages, in contrast to the observations reported by Chakrabarti et al (2020) and Chakrabarti et al (2021). One possibility is that rodent malaria parasites, as used in our study, may lack Apicortin in merozoites, whereas human malaria parasites do. However, transcriptomic and proteomic datasets on PlasmoDB (http://plasmodb.org) indicate that Apicortin levels are very low in the asexual blood stages of both rodent and human *Plasmodium* species.

Moreover, our previous work has shown that certain conoid-associated proteins exhibit stage-specific expression. For example, SAS6L is present in the invasive stages within the mosquito vector but undetectable during erythrocytic development (Wall et al, 2016). Similarly, in our study on conoid architecture (Koreny et al, 2021), we identified several parasite proteins expressed only in mosquito vector stages. These observations support the possibility that Apicortin may likewise exhibit stage-specific regulation and be largely absent from the blood-stage merozoites. Overall, the location of Apicortin implicates it in apical complex organization throughout parasite transmission stages within the mosquito.

A microtubule-associated role for proteins that contain TPPP domains (also referred to as p25alpha domains) has been proposed in a number of physiological contexts in humans, including a role in myelination of oligodendrocytes (Fu et al, 2019), and linked to the neurodegenerative features of both Alzheimer's and Parkinson's diseases (Olah et al, 2011). DC domains were first characterized in human doublecortin (DCX), a protein required for neuronal migration during brain development. Doublecortin and related doublecortin-like kinases are conserved in metazoa and are characterized by tandem DC domains (N-DC and C-DC), both of which can bind MTs and contribute to a range of microtubule-related functions (Reiner et al, 2006).

The distinct domain arrangement and modest sequence identity of Apicortins even in the structurally well-predicted DC domain argue that these proteins be considered as unique. This distinction is further supported by the specialized fibrillar arrangement of tubulin within the apicomplexan conoid complex—referred to as conoid fibrils or the conoid canopy—which is highly distorted compared with cellular MTs (Hu et al, 2002; Sun et al, 2022). Strikingly, both TPPP and DC domains bind to MTs between the protofilaments from which they—and the conoid fibrils—are built (Moores et al, 2004; Legal et al, 2025). This inter-protofilament binding site differs subtly in MTs with differing protofilament numbers (Moores et al, 2004), and may be dramatically modified by specific binding proteins, including Apicortin, to form the conoid complex.

PbApicortin expression was not detected in blood-stage merozoites, consistent with the absence of other conoid-associated proteins that are restricted to ookinetes and sporozoites (Wall et al, 2016; Koreny et al, 2021). This stage-specific restriction suggests that Apicortin contributes to the structural or functional

properties of the apical complex that are uniquely required for parasite development within the mosquito vector, but dispensable during erythrocyte invasion. In contrast, some conoid-associated proteins are expressed across all three invasive stages: ookinetes, sporozoites, and merozoites (Koreny et al, 2021), highlighting the idea that although there is a conserved apical complex scaffold, the composition may be remodelled to meet the distinct demands of different life-cycle stages. For example, merozoites invade host cells, but both ookinetes and sporozoites must invade and traverse host tissue.

Genetic deletion of *P. berghei* Apicortin had no detectable consequence for parasite differentiation, proliferation, or transmission through the mosquito vector. This may be because Apicortin function is redundant for conoid formation or function, or because conoid disruption is insufficient to prevent *P. berghei* transmission.

Overall, this study demonstrates that Apicortin is a component of the apical complex in both ookinetes and sporozoites, highlighting the conservation of conoid-related elements within *Plasmodium*. Despite its defined localization and MT-binding domain configuration, deletion of the *apicortin* gene does not impair ookinete or sporozoite morphology and motility, demonstrating that Apicortin is dispensable for parasite development and transmission.

# Materials and Methods

### Ethics statement

The animal work performed in this study has passed an ethical review process and was approved by the United Kingdom Home Office. Work was carried out under UK Home Office Project Licenses (PDD2D5182; PP3589958) in accordance with the UK "Animals (Scientific Procedures) Act 1986." Six- to eight-week-old female CD1 outbred mice from Charles River Laboratories were used for all experiments. The mice were kept in a 12-h light and 12-h dark (7 AM until 7 PM) cycle, at a temperature between 20°C and 24°C and the humidity between 40% and 60%.

### Generation of transgenic parasites

To generate the GFP-tagged lines, a region of *PbApicortin* downstream of the ATG start codon was amplified, ligated to p277 vector, and transfected as described previously (Guttery et al, 2012). The p277 vector contains the human *dhfr* cassette, conveying resistance to pyrimethamine. Schematic representations of the endogenous gene locus, the constructs, and the recombined gene locus can be found in Fig S1A. Diagnostic PCR was used with primer 1 (IntT275) and primer 3 (ol492) to confirm integration of the GFP targeting construct (Fig S1B).

WT-GFP sporozoites isolated from salivary glands. The arrow indicates the apical end of sporozoites. Scale bar = 5 $\mu$m. **(K)** Quantitative data for motile sporozoites from salivary glands for *ΔPbApicortin* and WT-GFP lines based on two independent experiments. **(L)** Bite-back experiments show successful transmission of *ΔPbApicortin* parasites (black bar) from mosquito to mice, like WT-GFP parasites (white bar). Mean ± SD; n = 3 independent experiments. Source data are available for this figure.

To delete the gene for Apicortin, the targeting vector for *apicortin was* constructed using the pBS-DHFR plasmid, which contains polylinker sites flanking a *T. gondii dhfr/ts* expression cassette conferring resistance to pyrimethamine, as described previously (Tewari et al, 2010). The 5′ upstream sequence of *apicortin* was amplified from genomic DNA and inserted into *Apa*I and *Hind*III restriction sites upstream of the *dhfr/ts* cassette of pBS-DHFR. A DNA fragment amplified from the 3′ flanking region of *apicortin* was then inserted downstream of the *dhfr/ts* cassette using *Eco*RI and *Xba*I restriction sites. The linear targeting sequence was released from the plasmid using *Apa*I/*Xba*I restriction enzymes. A schematic representation of the endogenous *apicortin* locus, the construct, and the recombined locus can be found in Fig S1C. A diagnostic PCR with primer 1 (IntN141_5) and primer 2 (ol248) was used to confirm integration of the targeting construct, and primer 3 (KO1) and primer 4 (KO2) were used to confirm deletion of the *apicortin* gene (Fig S1D). A list of primers used to amplify gene sequences can be found in Table S1.

*P. berghei* ANKA line 2.34 (for GFP-tagging) or ANKA line 507cl1 expressing GFP (for gene deletion) parasites were transfected by electroporation (Janse et al, 2006).

### Live-cell imaging

Different stages of parasite development during the asexual blood stage, the zygote-to-ookinete transformation, and oocyst development were analysed for PbApicortin-GFP expression and location using 10x and 63x oil immersion objectives on a Zeiss Axio Imager M2 microscope and analysed with AxioVision 4.8.2 and Zen lite 3.8 (Zeiss) software.

### Ookinete culture and purification

Mouse blood infected with PbApicortin-GFP parasites, with ~10% gametocytaemia, was incubated in ookinete culture medium (RPMI 1640+20% FBS+ 100 *µ*M xanthurenic acid) for 24 h. Under these conditions, the gametocytes differentiate to gametes, fertilize, and produce zygotes within 30 min. The zygotes differentiate to mature ookinetes over 24 h. Developing ookinetes (at 6–7 h post-incubation) and mature ookinetes (at 24 h) were purified using magnetic beads coated with antibodies against P28 (a surface protein of zygotes and ookinetes). The purified ookinetes were fixed with 4% PFA and used for structured illumination and expansion microscopy (Zeeshan et al, 2022).

### Structured illumination microscopy (3D-SIM)

The fixed ookinetes were resuspended in RPMI 1640 containing Hoechst dye and P28 antibody labelled with Cy3. Cells were scanned with an inverted microscope using Zeiss Plan-Apochromat 63x/1.4 oil immersion or Zeiss C-Apochromat 63×/1.2 W Korr M27 water immersion objective on a Zeiss Elyra PS.1 microscope, using the structured illumination microscopy (3D-SIM) technique. The correction collar of the objective was set to 0.17 for optimum contrast. The following settings were used in SIM mode: lasers, 405 nm: 20%, 488 nm: 16%, 561 nm: 8%; exposure times 200 ms (Hoechst), 100 ms (GFP for Apicortin), and 200 ms (dxRed for P28); three grid rotations, five phases. The band-pass filters BP 420–480 + LP 750, BP 495–550 + LP 750, and BP 570–620 + 750 were used for the blue, green, and red channels, respectively. Multiple focal planes (Z-stacks) were recorded with 0.2-*µ*m step size; later post-processing, a Z correction was done digitally on the 3D rendered images to reduce the effect of spherical aberration (reducing the elongated view in Z; a process previously tested with 0.1-µm fluorescent beads, T7284; Thermo Fisher Scientific). Registration correction was applied based on control images of multicoloured fluorescent beads. Images were processed, and all focal planes were digitally merged into a single plane (maximum intensity projection). The images recorded in multiple focal planes (Z-stack) were 3D rendered into virtual models and exported as images and movies (see Supplementary Material). Processing and export of images and videos were done by Zeiss Zen 2012 Black edition, Service Pack 5, and Zeiss Zen 2.1 Blue edition.

### Ultrastructure expansion microscopy (U-ExM)

Fixed ookinetes were sedimented onto 10-mm round poly-D-lysine–coated coverslips (A3890401; Gibco) for 30 min, then prepared for U-ExM as described previously (Gambarotto et al, 2021; Zeeshan et al, 2022). Expanded gels were immunolabelled using primary antibodies against *α*-tubulin (mouse, 1:1,000 dilution; Sigma-Aldrich) and anti-GFP (rabbit, 1:200) in 3% BSA in PBS overnight. The gels were washed three times for 10 min with PBST (PBS with 0.1% Tween-20) and then incubated with secondary antibodies anti-mouse Alexa 488 (1:100 dilution; Thermo Fisher Scientific) and anti-rabbit Alexa 568 (1:1,000) in PBS. The gels were washed again and expanded overnight. Images were acquired with an inverted microscope using Zeiss Plan-Apochromat 63x/1.4 oil immersion or Zeiss C-Apochromat 63×/1.2 W Korr M27 water immersion objective on a Zeiss Elyra PS.1 microscope.

### Phenotypic analyses

To study the phenotype of *ΔPbApicortin*, ~60,000 parasites of either the *ΔPbApicortin* or WT-GFP lines were injected intraperitoneally (i.p.) into mice. Asexual blood stages and gametocyte production were monitored by microscopy on Giemsa-stained thin smears. Four to five days post-infection, exflagellation and ookinete conversion were examined as described previously (Guttery et al, 2012) using a Zeiss Axio Imager M2 microscope (Carl Zeiss, Inc.) fitted with an AxioCam ICc1 digital camera. To analyse mosquito transmission of *ΔPbApicortin* and WT-GFP parasites, 50–60 *A. stephensi* SD 500 mosquitoes were allowed to feed for 20 min on anaesthetized, infected mice with a ~15% asexual parasitaemia and carrying comparable numbers of gametocytes as determined on Giemsa-stained blood films. To assess mid-gut infection, ~15 guts were dissected from mosquitoes on day 14 post-feeding and oocysts were counted on a Zeiss Axio Imager M2 microscope using 10x and 63x oil immersion objectives. On day 21 post-feeding, another 20 mosquitoes were dissected, and their guts and salivary glands crushed separately in a loosely fitting homogenizer to release sporozoites, which were then quantified using a haemocytometer or used for imaging and motility assays. Mosquito bite-back experiments were performed 21 d post-feeding using naive

mice; 15–20 mosquitoes infected with either WT-GFP or *ΔPbApicortin* parasites were fed for at least 20 min on naive CD1 outbred mice, and then, infection was monitored after 3 d by examining a blood smear stained with Giemsa's reagent. For comparison between *ΔPbApicortin* and WT-GFP, an unpaired *t* test was used.

## Ookinete motility assays

The motility of *ΔPbApicortin* ookinetes was assessed using Matrigel as described previously (Volkmann et al, 2012; Zeeshan et al, 2020). Briefly, ookinete cultures grown for 24 h were added to an equal volume of Matrigel (Corning), mixed thoroughly, dropped onto a slide, covered with a coverslip, and sealed with nail polish. The Matrigel was then allowed to set at 20°C for 30 min. After identifying a field containing an ookinete, time-lapse videos (one frame every 5 s for 100 cycles) were collected using the differential interference contrast settings with a 63× objective lens on a Zeiss Axio Imager M2 microscope fitted with an AxioCam ICc1 digital camera and analysed with Zen lite 3.8 software (Zeiss).

## Sporozoite motility assays

Sporozoites were isolated from salivary glands of mosquitoes infected with either WT-GFP or *ΔPbApicortin* parasites on day 21 post-infection. Isolated sporozoites in RPMI 1640 medium containing 3% bovine serum albumin (Thermo Fisher Scientific) were pelleted (5 min, 2,500*g*, 4°C) and used for motility assays as described previously (Wall et al, 2019). Briefly, a drop (6 *µ*l) of suspended sporozoites was transferred onto a microscope glass slide with a coverslip. Time-lapse videos of sporozoites (one frame every 1 s for 50 cycles) were taken using the differential interference contrast settings with a 63x objective lens on a Zeiss Axio Imager M2 microscope and analysed with Zen lite 3.8 software (Zeiss).

## Inhibitor assay

To investigate the influence of microtubule-modulating compounds on Apicortin-GFP localization and on the overall ookinete conversion efficiency of the *ΔPbApicortin* (KO) line, we carried out comparative drug treatment assays. To assess effects on apical localization, *Apicortin*-GFP parasites were collected in ookinete activation medium and activated for 1 h at 20°C. After activation, parasites were exposed to 1 or 10 *µ*M nocodazole or Taxol, and changes in Apicortin-GFP distribution were examined at 6 and 24 h post-incubation using live fluorescence imaging with a 63× objective lens on a Zeiss Axio Imager M2 microscope fitted with an AxioCam ICc1 digital camera and analysed with Zen lite 3.8 software (Zeiss). Representative images were acquired through structured illumination microscopy (SIM).

To evaluate the impact of microtubule-depolymerizing agents on overall ookinete conversion in the *ΔPbApicortin* line, a parallel assay was performed using WT-GFP parasites as controls. Parasites were collected in activation medium, activated as above, and treated with the same drug concentrations. Ookinete conversion efficiency was quantified 24 h post-treatment.

## Protein sequence analysis

Orthologues of Apicortin and doublecortin were identified using InterPro (Blum et al, 2025) and aligned using Clustal Omega hosted at EMBL-EBI (Madeira et al, 2024). The structure of *P. berghei* Apicortin was predicted using the AlphaFold structural database (Jumper et al, 2021; Varadi et al, 2022).

## Statistical analysis

All statistical analyses were performed using GraphPad Prism 9 (GraphPad Software).

# Data Availability

Source data are provided within this article as Supplementary Source Data Files.

# Supplementary Information

# Acknowledgements

We thank Julie Rodgers for helping in maintaining the insectary and other technical works. This work was supported by an ERC advanced grant funded by UKRI Frontier Science (EP/XO247761); the MRC UK (MR/K011782/1) and BBSRC (BB/N017609/1) to R Tewari; the BBSRC (BB/N017609/1) to M Zeeshan; the BBSRC (BB/N018176/1) to C Moores; and the Francis Crick Institute (FC001097), which receives funding from Cancer Research UK (FC001097), the UK Medical Research Council (FC001097), and the Wellcome Trust (FC001097), to AA Holder. For Open Access, the authors have applied a CC BY public copyright licence to any Author Accepted Manuscript version arising from this submission.

## Author Contributions

M Zeeshan: formal analysis, validation, investigation, visualization, methodology, and writing—original draft, review, and editing.
A Mishra: formal analysis, validation, investigation, visualization, and methodology.
SL Pashley: formal analysis, validation, investigation, visualization, and methodology.
R Markus: software, validation, and visualization.
D Brady: formal analysis, validation, and methodology.
AA Holder: conceptualization, supervision, funding acquisition, visualization, and writing—review and editing.
C Moores: conceptualization, data curation, formal analysis, funding acquisition, and writing—review and editing.
R Tewari: conceptualization, resources, data curation, formal analysis, supervision, funding acquisition, validation, investigation, visualization, methodology, and writing—original draft, review, and editing.

## Conflict of Interest Statement

The authors declare that they have no conflict of interest.

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
