## [Reviewer comments · Life Science Alliance]

Apicortin defines the Plasmodium apical conoid body but is dispensable for the parasite life cycle

Mohammad Zeeshan, Akancha Mishra, Sarah Pashley, Robert Markus, Declan Brady, Anthony Holder, Carolyn Moores, and Rita Tewari

DOI: <https://doi.org/10.26508/lsa.202503522>

Corresponding author(s): Rita Tewari, University of Nottingham

Review Timeline:

Submission Date:	2025-10-07
Editorial Decision:	2025-11-03
Revision Received:	2025-12-10
Editorial Decision:	2025-12-22
Revision Received:	2025-12-30
Accepted:	2026-01-05

Scientific Editor: Tim Fessenden

Transaction Report:

November 3, 2025

Re: Life Science Alliance manuscript #LSA-2025-03522

Prof. Rita Tewari
University of Nottingham
School of Life Sciences
Queens Medical Centre
Nottingham NG7 2UH
United Kingdom

Dear Dr. Tewari,

Thank you for submitting your manuscript entitled "Apicortin defines the Plasmodium apical conoid body during transmission but is dispensable for the parasite life cycle" to Life Science Alliance. The manuscript was assessed by expert reviewers, whose comments are appended to this letter.

As you will see, reviewers were unanimous in their support for these findings on Apicortin and the conoid body in P berghi. Reviewers requested a discussion of discordant results previously reported by Chakrabarti et al, and made various suggestions to improve this work. Reviewer 1 suggested two ways to strengthen and extend these insights in Apicortin. We feel these additions would improve this work and we encourage you to include them in a revision, if possible, however we leave this to your discretion.

I would be happy to discuss the revision in more detail via email or phone/videoconferencing if helpful. Please let me know which option you prefer, if any.

While you are revising your manuscript, please also attend to the below editorial points to help expedite the publication of your manuscript. Please direct any editorial questions to the journal office. When submitting the revision, please include a letter addressing the reviewers' comments point by point.

Thank you for this interesting contribution to Life Science Alliance. We hope that the comments below will prove constructive as your work progresses, and we are looking forward to receiving your revised manuscript.

Sincerely,

-- Summary blurb (enter in submission system): A short text summarizing in a single sentence the study (max. 200 characters including spaces). This text is used in conjunction with the titles of papers, hence should be informative and complementary to

the title and running title. It should describe the context and significance of the findings for a general readership; it should be written in the present tense and refer to the work in the third person. Author names should not be mentioned.

B. MANUSCRIPT ORGANIZATION AND FORMATTING:

Reviewer #1 (Comments to the Authors (Required)):

Summary of the Study

This manuscript examines the role of Apicortin, a conserved protein in the apical complex, in Plasmodium ookinetes and sporozoites. Using gene knockout (KO) and live-cell imaging techniques, the authors explore whether Apicortin helps organize the parasite's conoid-like structure. Through fluorescence microscopy of tagged lines (Apicortin-GFP and EB1-mCherry), they reveal distinct apical localization patterns across different developmental stages. Ultrastructure Expansion Microscopy (U-ExM) was employed to evaluate apical polarity and microtubule organization in the absence of Apicortin. Although Apicortin localizes to the apical region, its deletion does not visibly change microtubule arrangement, leading the authors to conclude it is likely non-essential for maintaining conoid structure under normal conditions. The discussion broadens these findings to the wider context of conoid function and possible compensatory mechanisms.

Overall Evaluation

The study is thorough, well-written, and methodologically robust. The microscopy data are clear, and the use of U-ExM provides valuable structural detail. However, the final part of the discussion slightly overstates the functional implications of the findings, particularly the claim that Apicortin is a "structural component" of the apical complex contributing to "unique microtubule arrangements." Since microtubule organization and conoid morphology appear unchanged in the knockout, Apicortin seems non-essential rather than structural in this context. The authors could moderate their interpretation or include functional evidence to support it, for instance, testing whether the Apicortin KO shows increased sensitivity to microtubule destabilizers such as nocodazole or oryzalin.

Specific Comments

1. Conoid-Associated Proteins and Double Tagging

Known conoid-associated proteins in Plasmodium include Apicortin itself and End-Binding Protein 1 (EB1), among others (Wall et al., 2016; Koreny et al., 2021, 2022). The authors used parasites double-tagged with Apicortin-GFP and EB1-mCherry to compare their relative apical localization. While this provides useful spatial information, EB1 becomes more diffuse in mature stages, whereas Apicortin remains as a focal apical ring.

However, EB1 localization is not shown in the Apicortin knockout, which is a missed opportunity. Examining EB1 in the KO would clarify whether the absence of Apicortin affects EB1 recruitment or microtubule plus-end organization at the apical complex. Including such data would directly test whether Apicortin exerts any structural or regulatory influence on apical microtubules. Future work incorporating additional markers (e.g., SAS6L, RNG2, or other conserved apical complex proteins) could further refine the spatial map of Apicortin within the apical complex.

2. Figure 4: U-ExM and NHS Ester Staining

Figure 4E shows U-ExM images of WT and Apicortin KO parasites stained with anti-tubulin and NHS ester. Although the text claims that apical polarity and microtubule organization seem unchanged, the supporting evidence is limited. The only markers displayed are general stains (tubulin and NHS ester), which do not fully resolve polarity or conoid structure.

Furthermore, visual inspection indicates that the WT parasite shows denser NHS-ester-positive granules at the apical end compared to the KO, which may correspond to micronemes or other apical vesicles. This suggests subtle polarity or trafficking differences rather than the absence of changes. Quantitative analysis, such as counting granules or measuring signal intensity, would be required to determine if these differences are statistically significant. Co-staining with known micronemal, rhoptry, or IMC markers would help clarify the identity of these structures and evaluate whether Apicortin depletion indirectly affects apical organelle composition.

3. Additional Functional Insights

If feasible, examining EB1 localization in the KO and/or testing sensitivity to microtubule destabilization would directly determine whether Apicortin plays a role in microtubule stability or organization. These targeted functional assays would more effectively support the authors' interpretation than broader -omics approaches.

A proteomic comparison could be considered in future studies to explore compensatory pathways, but it is not essential for supporting the current conclusions.

Final Recommendations

The manuscript convincingly characterizes Apicortin localization and offers a solid framework for understanding its role in the Plasmodium apical complex. However, the conclusion that Apicortin is a "structural component" should be tempered, as the data presented are consistent with a non-essential or modulatory role. To strengthen the study within its current scope, the authors could:

- Quantify NHS-ester-positive granules to evaluate apical polarity;
- Include EB1 localization in the KO; and/or
- Test for sensitivity to microtubule destabilizers.

These additions would directly support the main claims and provide a more accurate picture of Apicortin's role in apical complex organization.

Reviewer #2 (Comments to the Authors (Required)):

It is a nice and carefully done work on the expression and function of apicortin in Plasmodium berghei. It has been shown that it is expressed at the apical end of invasive parasites only during development of transmission stages within the mosquito vector but not during asexual blood-stage replication, in both male and female gametocytes, and in flagellated male gametes. Deletion of the apicortin gene had no effect on parasite development and transmission through the mosquito. The article is an important contribution to a relatively neglected area of Plasmodium research. The experiments are well interpreted and their discussion is adequate.

Suggestions:

Species and genome names should be always in italic.

Line 101: I suggest citing here also Chakrabarti et al., 2020 (doi: 10.1242/dmm.042820.).

Line 179: 3D-SIM is resolved only in line 352.

Lines 276, 281: Koreny et al., 2022 - change for 2021

It seems that there are discrepancies between the authors' results (expression in blood stages; functional consequences) and those of Chakrabarti et al (2020, 2021) obtained with Plasmodium falciparum. It would be nice to read some discussing sentences.

References

Species and genome names should be in italic also in References.

Sun, S.Y., ... Proc Natl Acad Sci U S A 119. - The article number is missing: e2111661119

Legal et al. - Instead of bioRxiv, I suggest citing the final publication in Current Biology (<https://doi.org/10.1016/j.cub.2025.06.020>).

Reviewer #3 (Comments to the Authors (Required)):

Mohammad Zeeshan, Akancha Mishra, Sarah L Pashley, Robert Markus, Declan Brady, Anthony A. Holder, Carolyn Moores, Rita Tewari. Apicortin defines the Plasmodium apical conoid body during transmission but is dispensable for the parasite life cycle

Overall summary:

The protein apicortin was first defined in Toxoplasma (TgDCX) where it was shown to associate with the conoid, an organelle formed of tightly curved tubulin filaments. Loss of apicortin (gene KO) destabilizes the conoid structure and causes a severe lytic cycle defect. Heterologous expression of apicortin in vertebrate cells induces curvature upon binding to microtubules, suggesting that it plays a role in stabilizing conoid filaments. Apicortin homologs are found in other apicomplexans and in their free-living relatives Chromera and Vitrella.

This study uses a C-terminal knock-in GFP to endogenous apicortin to establish its expression pattern and to localize it in Plasmodium berghei, a rodent malaria used for its access to all life cycle stages. The authors demonstrate that apicortin is specifically expressed in the invasive stages present in the mosquito (ookinetes and sporozoites) but not the gametes, nor in the invasive stage present in the vertebrate host (merozoites). Targeted deletion of Apicortin does not affect development of gametes, oocyst numbers, or formation or motility of sporozoites.

This study has excellent microscopy and the negative data (lack of a phenotype during targeted deletion) is in contrast to the fitness-conferring role of apicortin in *Toxoplasma* tachyzoites. This is an important illustration of how the evolutionary trajectories of individual apicomplexan lineages differ. I have very few suggestions to improve this paper, but I would suggest that authors expand their discussion to address an apparent discrepancy in their results with a prior paper (Chakrabarti et al., 2021) that used *Plasmodium falciparum* to investigate apicortin and concluded that merozoites express this protein. The prior study uses antibody localization, with sera generated from bacterially expressed apicortin and tubulin proteins used for IFA microscopy. I have some concerns about the conclusions in the earlier paper, as fluorescence localization suggests an abundant signal in merozoites yet MS data deposited at PlasmoDB identifies only a single peptide, which is more in-line with observations in the current MS that conclude that expression is restricted to vector stage invasive forms. The MS would also benefit from a discussion of the group's prior findings for another conoid associated protein, SAS6L. Like apicortin, the conoid associated protein SAS6L is fitness conferring in *Toxoplasma*, shows restricted expression in *Plasmodium* ookinetes and sporozoites, yet is dispensable for these stages. Both proteins illustrate that while *Plasmodium* retains a vestigial conoid, this structure is less critical for invasion than the conoid in coccidians.

Suggested minor corrections:

71: "In these organisms" "In the latter organisms" would be clearer (specifying the non-apicomplexans)

89 "T. gondii host cell invasion" "T. gondii tachyzoite invasion of host cells" would specify the parasite stage as was defined for the *Plasmodium* stages.

107-9: "We demonstrate that Apicortin is expressed in the invasive stages present only within the mosquito vector." This sentence could be clarified as it is important. "We demonstrate that the mosquito vector-specific invasive forms (ookinetes and sporozoites) express apicortin whereas merozoites do not."

We thank the reviewers and the Editor for their constructive comments and valuable suggestions to improve our manuscript. We have addressed all comments and concerns as far as possible, and with additional new data. We hope that this revised manuscript will be acceptable for publication. Please find below detailed responses to all specific points raised by the reviewers.

Reviewer #1

Overall Evaluation

The study is thorough, well-written, and methodologically robust. The microscopy data are clear, and the use of U-ExM provides valuable structural detail. However, the final part of the discussion slightly overstates the functional implications of the findings, particularly the claim that Apicortin is a "structural component" of the apical complex contributing to "unique microtubule arrangements." Since microtubule organization and conoid morphology appear unchanged in the knockout, Apicortin seems non-essential rather than structural in this context. The authors could moderate their interpretation or include functional evidence to support it, for instance, testing whether the Apicortin KO shows increased sensitivity to microtubule destabilizers such as nocodazole or oryzalin.

Response: We appreciate the reviewer's positive and encouraging feedback. We agree that the potential functional role of Apicortin may have been overstated in the earlier version of the manuscript. As suggested, we examined the effect of nocodazole on Apicortin KO and WT parasites and saw no significant difference in either ookinete conversion rate or in ookinete morphology. We also assessed the effect of taxol (a microtubule-stabilizing agent) and observed defects in the morphological transformation of both WT and Apicortin KO ookinetes, but no differences, indicating that the Apicortin gene deletion had no effect in addition to that of the taxol treatment. These data are included in new supplementary Fig. S2D-F and we have moderated our interpretation accordingly. Please see line numbers 220-228 in the results section and 315-320 in the discussion.

Specific Comments

1. Conoid-Associated Proteins and Double Tagging

Known conoid-associated proteins in Plasmodium include Apicortin itself and End-

Binding Protein 1 (EB1), among others (Wall et al., 2016; Koreny et al., 2021, 2022). The authors used parasites double-tagged with Apicortin-GFP and EB1-mCherry to compare their relative apical localization. While this provides useful spatial information, EB1 becomes more diffuse in mature stages, whereas Apicortin remains as a focal apical ring.

However, EB1 localization is not shown in the Apicortin knockout, which is a missed opportunity. Examining EB1 in the KO would clarify whether the absence of Apicortin affects EB1 recruitment or microtubule plus-end organization at the apical complex. Including such data would directly test whether Apicortin exerts any structural or regulatory influence on apical microtubules.

Response: We thank the reviewer for this excellent suggestion to examine the effect of Apicortin gene deletion on the location of EB1. Unfortunately, we do not have an EB1-specific antibody to perform this analysis, but we will consider incorporating this approach into future studies.

Future work incorporating additional markers (e.g., SAS6L, RNG2, or other conserved apical complex proteins) could further refine the spatial map of Apicortin within the apical complex.

Response: We agree that it would be very informative to incorporate additional markers in future studies. However, we lack antibodies against these markers, which limits our ability to assess their location. We will include these markers once the necessary reagents become available.

2. Figure 4: U-ExM and NHS Ester Staining

Figure 4E shows U-ExM images of WT and Apicortin KO parasites stained with anti-tubulin and NHS ester. Although the text claims that apical polarity and microtubule organization seem unchanged, the supporting evidence is limited. The only markers displayed are general stains (tubulin and NHS ester), which do not fully resolve polarity or conoid structure.

Furthermore, visual inspection indicates that the WT parasite shows denser NHS-ester-positive granules at the apical end compared to the KO, which may correspond to micronemes or other apical vesicles. This suggests subtle polarity or trafficking differences rather than the absence of changes. Quantitative analysis, such as

counting granules or measuring signal intensity, would be required to determine if these differences are statistically significant.

Response: We have counted the number of granules and observed no significant difference between WT and Apicortin KO ookinetes. These quantification data are in supplementary Fig. S2C. Please see line numbers 220-221 in the result section.

Co-staining with known micronemal, rhoptry, or IMC markers would help clarify the identity of these structures and evaluate whether Apicortin depletion indirectly affects apical organelle composition.

Response: Unfortunately, we do not have antibodies against any of these markers, but we will include them in future studies once the necessary reagents become available.

3. Additional Functional Insights

If feasible, examining EB1 localization in the KO and/or testing sensitivity to microtubule destabilization would directly determine whether Apicortin plays a role in microtubule stability or organization. These targeted functional assays would more effectively support the authors' interpretation than broader -omics approaches.

Response: As stated above we cannot examine the effect of Apicortin gene deletion on the location of EB1 since we lack EB1-specific antibodies.

We have evaluated the effects of nocodazole and taxol treatment on the location of Apicortin in the ookinete stage using the Apicortin-GFP line. Nocodazole treatment had no effect on ookinete conversion, morphology, or the location of Apicortin-GFP. Taxol treatment impaired ookinete shape transformation but had no effect on the location of Apicortin-GFP at the apical end; the effect on ookinete morphology was similar in both Apicortin-GFP and Apicortin KO parasites. These observations suggest that the conoid is a relatively stable structure that was not substantially perturbed by these microtubule inhibitor treatments. These data are included in supplementary Fig. S1C-F. Please see line numbers 141-148 in the results section.

A proteomic comparison could be considered in future studies to explore compensatory pathways, but it is not essential for supporting the current conclusions.

Final Recommendations

The manuscript convincingly characterizes Apicortin localization and offers a solid framework for understanding its role in the Plasmodium apical complex. However, the conclusion that Apicortin is a "structural component" should be tempered, as the data presented are consistent with a non-essential or modulatory role. To strengthen the study within its current scope, the authors could:

- Quantify NHS-ester-positive granules to evaluate apical polarity;
- Include EB1 localization in the KO; and/or-
- Test for sensitivity to microtubule destabilizers-

These additions would directly support the main claims and provide a more accurate picture of Apicortin's role in apical complex organization.

Response: We thank the reviewer for their constructive comments. As far as is possible we have addressed these suggestions and added the new data into the revised manuscript.

Reviewer #2

It is a nice and carefully done work on the expression and function of apicortin in Plasmodium berghei. It has been shown that it is expressed at the apical end of invasive parasites only during development of transmission stages within the mosquito vector but not during asexual blood-stage replication, in both male and female gametocytes, and in flagellated male gametes. Deletion of the apicortin gene had no effect on parasite development and transmission through the mosquito. The article is an important contribution to a relatively neglected area of Plasmodium research. The experiments are well interpreted and their discussion is adequate.

Suggestions:

Species and genome names should be always in italic.

We have italicized all species and genus names as required.

Line 101: I suggest citing here also Chakrabarti et al., 2020 (doi: 10.1242/dmm.042820.).

Response: We have now cited this reference

Line 179: 3D-SIM is resolved only in line 352.

Response: We had already used 3D-SIM to resolve the apical end of ookinetes (Fig. 3A), which is mentioned at line number 172.

Lines 276, 281: Koreny et al., 2022 - change for 2021

Response: Thank you for the correction.

It seems that there are discrepancies between the authors' results (expression in blood stages; functional consequences) and those of Chakrabarti et al (2020, 2021) obtained with *Plasmodium falciparum*. It would be nice to read some discussing sentences.

Response: We agree with the reviewer. In our study, we observed no Apicortin expression in the asexual erythrocytic stages of *P. berghei*, contrary to the findings in *P. falciparum* reported by Chakrabarti et al. (2020, 2021). It is possible that rodent malaria parasites do not express Apicortin during these stages, whereas human malaria parasites do. However, transcriptomic and proteomic datasets available on PlasmoDB indicate that Apicortin expression is very low in the blood stages of both rodent and human malaria parasites. In addition, previously we observed that the conoid-associated protein SAS6L is expressed in the invasive stages within the mosquito vector and is undetectable during erythrocytic stages (Wall et al., 2016), supporting the possibility of stage-specific expression patterns. We have discussed this in line numbers 265-278.

References

Species and genome names should be in italic also in References.

Response: We have checked and italicized where required.

Sun, S.Y., ... Proc Natl Acad Sci U S A 119. - The article number is missing:
e2111661119

Response: We have added the article number for Sun et al.

Legal et al. - Instead of bioRxiv, I suggest citing the final publication in Current Biology (<https://doi.org/10.1016/j.cub.2025.06.020>).

Response: Done.

Reviewer #3

This study has excellent microscopy and the negative data (lack of a phenotype during targeted deletion) is in contrast to the fitness-conferring role of apicortin in *Toxoplasma* tachyzoites. This is an important illustration of how the evolutionary trajectories of individual apicomplexan lineages differ. I have very few suggestions to improve this paper, but I would suggest that authors expand their discussion to address an apparent discrepancy in their results with a prior paper (Chakrabarti et al., 2021) that used *Plasmodium falciparum* to investigate apicortin and concluded that merozoites express this protein. The prior study uses antibody localization, with sera generated from bacterially expressed apicortin and tubulin proteins used for IFA microscopy. I have some concerns about the conclusions in the earlier paper, as fluorescence localization suggests an abundant signal in merozoites yet MS data deposited at PlasmoDB identifies only a single peptide, which is more in-line with observations in the current MS that conclude that expression is restricted to vector stage invasive forms. The MS would also benefit from a discussion of the group's prior findings for another conoid associated protein, SAS6L. Like apicortin, the conoid associated protein SAS6L is fitness conferring in *Toxoplasma*, shows restricted expression in *Plasmodium* ookinetes and sporozoites, yet is dispensable for these stages. Both proteins illustrate that while *Plasmodium* retains a vestigial conoid, this structure is less critical for invasion than the conoid in coccidians.

Response: We appreciate the reviewer's critical analysis and thank them for their supportive comments.

Suggested minor corrections:

71: "In these organisms" "In the latter organisms" would be clearer (specifying the non-apicomplexans)

Response: We have revised the sentence to make it clearer.

89 "T. gondii host cell invasion" "T. gondii tachyzoite invasion of host cells" would specify the parasite stage as was defined for the Plasmodium stages.

Response: We have modified the sentence to clearly specify the stages of the life cycle.

107-9: "We demonstrate that Apicortin is expressed in the invasive stages present only within the mosquito vector." This sentence could be clarified as it is important. "We demonstrate that the mosquito vector-specific invasive forms (ookinetes and sporozoites) express apicortin whereas merozoites do not."

Response: We have revised the sentence to clarify the stages.

December 22, 2025

RE: Life Science Alliance Manuscript #LSA-2025-03522R

Prof. Rita Tewari
University of Nottingham
School of Life Sciences
Queens Medical Centre
Nottingham NG7 2UH
United Kingdom

Dear Dr. Tewari,

Thank you for submitting your revised manuscript entitled "Apicortin defines the Plasmodium apical conoid body but is dispensable for the parasite life cycle", which was returned to Reviewers 1 and 2 for re-review. As the reviewers are now satisfied, we would be happy to publish your paper in Life Science Alliance pending final revisions necessary to meet our formatting guidelines. We appreciate your patience in waiting to learn our decision.

- Please add a Summary Blurb/Alternate Abstract in our system.
- We suggest slightly editing the Abstract to remark on Apicortin's known role in microtubule binding and stabilization before, and not after, the statement "How these structures are maintained is poorly understood, but it may involve Apicortin..." However we leave making this change to your discretion.
- Please upload a clean manuscript without the track changes. The version with the changes you may upload with the file designation "Related manuscript file."
- A "Data Availability" section should be placed after the Materials & Methods section. Please consult our guidelines at <https://www.life-science-alliance.org/manuscript-prep#format>
- Please add an Author Contributions section to your main manuscript text.
- The contributions selected for Anthony Holder do not qualify them for authorship. Please either update the contributions in our system and in the Author Contributions section of the manuscript, or let us know if the author needs to be removed (and added eventually to the acknowledgment section).
- Please add a Conflict of Interest statement to your main manuscript text.
- Please move your main, supplementary figure, table, and video legends in the main manuscript text after the references section.

A. FINAL FILES:

- An editable version of the final text (.DOC or .DOCX) is needed for copyediting (no PDFs).
- High-resolution figure, supplementary figure and video files uploaded as individual files: See our detailed guidelines for preparing your production-ready images, <https://www.life-science-alliance.org/authors>
- Summary blurb (enter in submission system): A short text summarizing in a single sentence the study (max. 200 characters including spaces). This text is used in conjunction with the titles of papers, hence should be informative and complementary to

the title. It should describe the context and significance of the findings for a general readership; it should be written in the present tense and refer to the work in the third person. Author names should not be mentioned.

B. MANUSCRIPT ORGANIZATION AND FORMATTING:

Thank you for your attention to these final processing requirements. Please revise and format the manuscript and upload materials as soon as you are able.

Sincerely,

Reviewer #1 (Comments to the Authors (Required)):

I thank the authors for providing thoughtful responses to my comments, and I have no further comments. Congratulations on this beautiful study.

Reviewer #2 (Comments to the Authors (Required)):

It is a revised version. My suggestions were accepted and questions were answered. I suggest publishing this paper.

January 5, 2026

RE: Life Science Alliance Manuscript #LSA-2025-03522RR

Prof. Rita Tewari
University of Nottingham
School of Life Sciences
Queens Medical Centre
Nottingham NG7 2UH
United Kingdom

Dear Dr. Tewari,

Thank you for submitting your Research Article entitled "Apicortin defines the Plasmodium apical conoid body but is dispensable for the parasite life cycle". It is a pleasure to let you know that your manuscript is now accepted for publication in Life Science Alliance. Congratulations on this interesting work.

DISTRIBUTION OF MATERIALS:

Again, congratulations on a very nice paper. I hope you found the review process to be constructive and are pleased with how the manuscript was handled editorially. We look forward to future exciting submissions from your lab.

Sincerely,
